# communications
# engineering

## COMMENT

# The 2023 Kahramanmaraș Earthquake Sequence: finding a path to a more resilient, sustainable, and equitable society

Carmine Galasso [1✉] & Eyitayo A. Opabola[2]

Learning from the 2023 Kahramanmaraș Earthquake Sequence offers valuable insights into disaster recovery. Here we delve into the intricacies of the "Build Back Better" (BBB) concept, underscoring the importance of recovery and reconstruction efforts toward a future that is not only more resilient but also more sustainable and equitable.

A moment magnitude ($M_w$) 7.8 earthquake, with an epicenter located at 37.226°N, 37.014°E[1], occurred in the early hours of February 6, 2023. This powerful and shallow event caused widespread damage and casualties in southeastern Turkey and northern Syria, affecting an area of about 350,000 square kilometers, or the size of Germany. This earthquake was followed by an $M_w$ 7.5 event approximately nine hours after the first event. The total number of affected people in the Turkish and Syrian regions was about 14 million and 9 million, respectively[2,3]. A total death toll of about 60,000 (50,783 in Turkey[4] and over 7000 in Syria[5]) has been reported. Over 500,000 buildings suffered heavy damage or collapsed in Turkey alone[2]. About three million people in Turkey still lived in tents three months after the sequence[6]. Figure 1 shows the impact (i.e., fatalities and displaced people—defined as the number of people whose houses were destroyed or heavily damaged) of the 2023 Kahramanmaraș Earthquake Sequence (considering the two main events mentioned above) relative to other seismic events between 2001 and 2023 from the International Disaster Database (EM-DAT)[7]. We also compare the number of fatalities and displaced people (depicted by the size of the colored circles) in Turkey and Syria with seven other seismic events in low- and lower-middle-income countries since 2001. We note that Turkey and Syria are classified as upper-middle- and low-income countries, respectively[8]. As shown in Fig. 1, the 2023 Kahramanmaraș Earthquake Sequence is the fifth deadliest earthquake in the 21st century, with the third most severe earthquake-induced people displacement.

According to various earthquake reconnaissance efforts[9,10], many buildings that collapsed or were severely damaged—causing casualties and displacements—were either old, poorly constructed, or not compliant with modern seismic codes. This was due to corruption, lack of awareness, or profit motive by multifamily residential building owners or contractors wanting unauthorized extra floors or expanded balconies to maximize profit[11]. Over seven million existing buildings in Turkey benefit from the Turkish government's construction amnesty policy, which provides permits for non-code-compliant buildings[12]. About 300,000 buildings across the affected region benefited from the construction amnesty[11], generating $4.2bn in government revenue. In Syria, the earthquake exacerbated the effects of the ongoing war, which had already destroyed or damaged many buildings and infrastructure. The lack of resources, maintenance, and regulation made many structures unsafe and unstable. The earthquake also triggered

[1]Department of Civil, Environmental, and Geomatic Engineering, University College London, Gower St, London WC1E 6BT, UK. [2]Department of Civil and Environmental Engineering, University of California, 775 Davis Hall, Berkeley, CA 94720, USA. ✉email: c.galasso@ucl.ac.uk

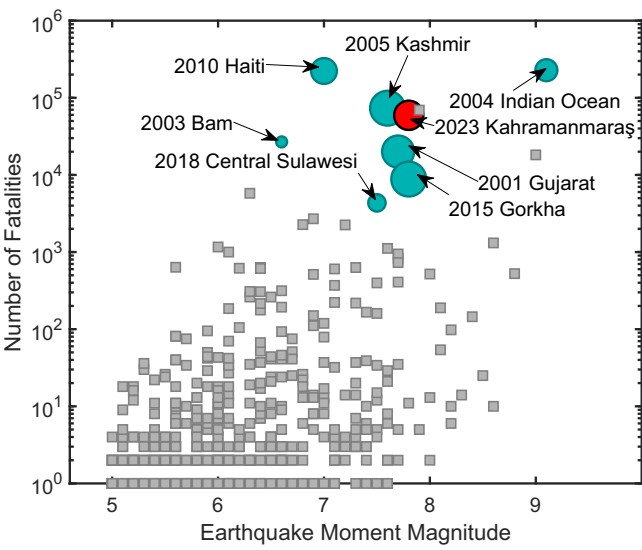

**Fig. 1 Impact of earthquakes on countries in the last two decades.** Relationship between earthquake moment magnitude ($M_w$) and number of fatalities globally in the last two decades. We highlight seven seismic events in low- and lower-middle-income countries that have experienced earthquakes with $M_w > 6.5$ and a number of fatalities ≥4000 with green circles together with the 2023 Turkey-Syria earthquake (red circle). The size of the circles represents the number of displaced people following the eight highlighted events. The gray square markers depict events with a number of fatalities <4000 or not in the considered country income class. Data were derived from the International Disaster Database (EM-DAT)[7]. The 2018 Central Sulawesi and 2004 Indian Ocean earthquakes triggered tsunamis.

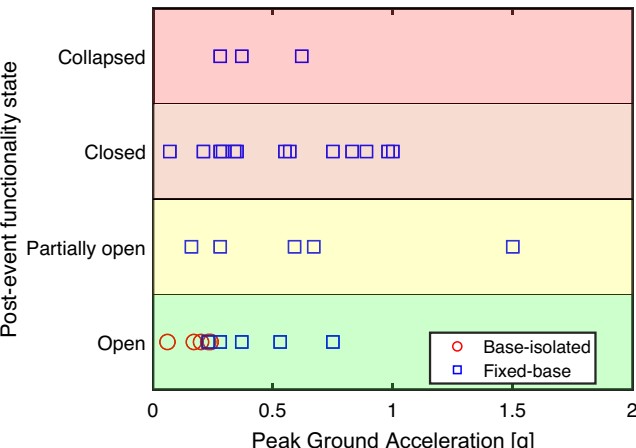

**Fig. 2 Impact of seismic isolation systems on post-event functionality of hospitals.** Post-event functionality states of 34 hospital buildings in the affected region relative to the observed peak ground accelerations (PGA) derived from the initial USGS shake map in units of $g$, a measure of acceleration caused by gravity (an object at rest on Earth's surface is subject to 1g, equaling the conventional value of gravitational acceleration on Earth, about 9.8 m/s²). The hospitals were inspected by the United States Earthquake Engineering Research Institute (EERI) team[10] and United Kingdom's Earthquake Engineering Field Investigation Team (EEFIT)[40] about six weeks after the disaster. There are five base-isolated hospital buildings in the affected region, with another four (not shown in the figure) under construction when the earthquake occurred.

landslides and liquefaction, further compromising various civil infrastructure systems' foundations and integrity.

## Successes in risk reduction measures are often invisible, resulting in a lack of incentives for proactive decision-making

Disaster risk reduction is critical to prevent losses and other impacts from future disasters. However, its success often remains obscured due to the inherent challenges of evaluating outcomes and recognizing effective interventions, for instance, because hazard events do not occur in a given region/time window. Furthermore, a typical scenario where successful risk reduction remains invisible is when significant hazard events, such as the 2023 Kahramanmaraş Earthquake Sequence, cause catastrophic impacts, diverting attention away from past mitigation efforts[13,14]. In such cases, post-disaster analysis rarely revisits previous interventions to assess their effectiveness. Hence, highlighting the benefits of successful risk reduction becomes essential. Celebrating past successes can help sustain and amplify ongoing disaster risk reduction efforts and provide positive examples to learn from, rather than focusing only on adverse events and facts that often dominate the news and research, as in the case of the 2023 Kahramanmaraş Earthquake Sequence.

For instance, despite the widespread damage to buildings and infrastructure across the affected provinces, an example of success stories is the relatively good performance of tunnel-form reinforced concrete (RC) buildings[9,10]. The design and construction methods for tunnel-form buildings (i.e., shear walls cast simultaneously with slabs using box-shaped formwork) result in stiffer and stronger lateral load-resisting systems, significantly reducing the collapse risk of this building typology[15]. Residential tunnel-form buildings constructed through the government-backed mass housing development program (i.e., Housing Development Administration of the Republic of Türkiye, TOKİ)

maintained occupancy following the earthquake. According to TOKİ, about 133,759 TOKİ-built buildings across the ten affected provinces did not suffer structural damage[16]. Tunnel-form buildings also performed well following the 1999 $M_w$ 7.6 Izmit earthquake in Turkey, with no reported cases of severe damage or collapse[15].

The performance of base-isolated hospital buildings represents another example of success stories during the 2023 Kahramanmaraş Earthquake Sequence. In 2013, the Turkish Ministry of Health announced a new policy requiring base isolation for hospital buildings with bed capacity exceeding 100 in earthquake-prone zones[17]. According to the technical specifications prepared by the Ministry, base-isolated hospitals must achieve continued functionality and immediate occupancy following design level and maximum considered earthquake events, respectively. The 2023 Kahramanmaraş Earthquake Sequence highlighted this policy's positive significance and further demonstrated the efficacy of seismic isolation systems in achieving the Turkish Ministry of Health's desired performance objectives for hospitals. Five base-isolated hospital buildings within the affected regions remained functional following the event. On the other hand, several conventional hospital buildings (i.e., with fixed bases) suffered partial or total damage (Fig. 2).

Furthermore, the Turkish government partnered with the World Bank and its Global Facility for Disaster Reduction and Recovery (GFDRR) in an effort to ensure that Turkish schools are safer and more disaster-resilient[18]. Since 2017, this partnership has resulted in the design and construction of 24 seismic-resistant schools in the regions affected by the 2023 Kahramanmaraş Earthquake Sequence. All of these 24 schools survived the earthquake sequence without damage[19]. Some of these schools have served as hubs to provide vital services to disaster victims[19].

These inspiring stories show how some buildings and infrastructure performed well during the 2023 Kahramanmaraş Earthquake Sequence. Furthermore, these stories provide good templates for a more resilient future.

Indeed, in the aftermath of the disaster, Turkey and Syria launched recovery efforts with the help of international and local partners. However, these questions remain: how can they truly "Build Back Better"? How can they reduce the risk of future disasters and shocks by improving their communities' and nations' resilience and sustainability? How can they address the underlying causes of vulnerability and inequality that make some communities more susceptible to harm? How can they seize the opportunity to transform their development pathways toward more resilient, sustainable, and inclusive outcomes?

## An opportunity to "Build Back Better"

The first challenge after a significant disaster (e.g., earthquake-induced disasters such as the 2023 Kahramanmaraş Earthquake Sequence) is to address immediate humanitarian needs. Lives must be saved, and necessities such as food, water, shelter, and medical care must be provided promptly. However, as the dust settles and the immediate crisis eases, attention turns to the monumental task of rebuilding. At this stage, the "Build Back Better" (BBB) approach[20] comes into play, guiding decisions, policies, and actions that will shape the trajectory of the affected regions for years to come.

Indeed, future earthquakes are still natural phenomena, but the choices made during reconstruction can exacerbate or mitigate future seismic events' physical, environmental, and human impacts, making them future disasters. BBB is one of the four priorities for action (the fourth) in the 2015-2030 Sendai Framework for Disaster Risk Reduction[20]. The BBB concept became popular in 2006 during the large-scale reconstruction effort following the Indian Ocean Tsunami disaster of 2004. Ten key principles proposed by former US President Bill Clinton were adopted after that disaster. These principles included a commitment to community-led recovery, promoting fairness and equity, and leaving communities safer by reducing risks and building resilience. Before BBB, post-disaster reconstruction often consisted of simply repairing the physical damage a disaster had induced. However, rebuilding the built environment and infrastructure exactly as they were prior to a disaster often re-created the same vulnerabilities that existed earlier[21].

The United Nations Office for Disaster Risk Reduction (UNDRR) defines BBB as "The use of the recovery, rehabilitation, and reconstruction phases after a disaster to increase the resilience of nations and communities through integrating disaster risk reduction measures into the restoration of physical infrastructure and societal systems, and into the revitalization of livelihoods, economies, and the environment."

Based on this definition, the fundamental principle of BBB is resilience. This entails aiming at physical infrastructure, systems, and communities that "resist, absorb, accommodate, adapt to, transform and recover from the effects of a hazard in a timely and efficient manner, including through the preservation and restoration of its essential basic structures and functions through risk management." (UNDRR). It should involve using (cutting-edge) hazard-resistant technologies, materials, and building codes/practices incorporating the latest research and experiences, as well as strengthening a community's social fabric and networks and its disaster preparedness[21]. Local leaders and government agencies should work with community members to develop comprehensive disaster response plans. These plans ensure that everyone knows their role in times of crisis, reducing chaos and maximizing the effectiveness of relief efforts.

In an era of climate change, BBB should not stop at resilience; it should also embrace sustainability. As reconstruction proceeds, it is essential to integrate sustainable practices into every aspect of the rebuilding process. Sustainable building materials, energy-efficient designs, and incorporating renewable energy sources (when possible) all reduce the rebuilt structures' carbon footprint. Green spaces, parks, and urban gardens beautify the landscape, contribute to air quality, and provide recreation and social interaction spaces. Sustainable recovery can also consist of rethinking transportation systems to reduce emissions, promote cleaner transportation modes, and implement sustainable waste management and recycling systems. Moreover, it involves incorporating nature-based solutions that restore and enhance natural ecosystems that can mitigate the impact of natural hazards.

Furthermore, BBB goes beyond just the physical environment; it should also address social equity through a "people-centered" lens[22]. In fact, the BBB concept is often too vague to provide clear guidance for housing reconstruction and broader recovery. It is frequently used in a limited sense to denote safer construction or engineering risk reduction without a comprehensive understanding of what constitutes a "better" life for affected individuals, particularly those vulnerable and marginalized. Disasters often reveal and exacerbate existing inequalities, disproportionately affecting vulnerable populations[23,24]. It is crucial to ensure that the reconstruction efforts prioritize the needs of well-informed and empowered residents in decision-making and construction, promoting inclusivity and reducing disparities. For instance, affordable housing could be a critical element in this regard. Rebuilding homes that are accessible and affordable for all income levels ensures that the entire community can benefit from the reconstruction. Land-use policies prioritizing affordable housing and preventing the displacement of low-income residents should be integrated into the reconstruction plan. Education is another pillar of social equity. Schools are not just buildings but the foundation of a thriving community. Rebuilding schools that are safer and more resilient, more accessible, and well-equipped empowers the next generation and ensures that educational opportunities are not lost due to the disaster. Healthcare facilities should also be a focus. An earthquake often strains the existing healthcare infrastructure, making rebuilding and expanding healthcare services essential. Accessibility to quality healthcare is a fundamental right, and post-disaster reconstruction could offer the chance to strengthen healthcare systems for the entire community. Employment and economic recovery are intertwined with social equity. Providing job opportunities for residents not only aids the recovery process but also empowers the community. To facilitate economic recovery, it is necessary to provide financial aid, offer training, and support the rebuilding of businesses[21].

These are just possible examples (based on our experience on some recent projects in the Global South) of specific programs that could be incorporated into a BBB strategy but also in more general frameworks for risk-informed urban planning, design, and decision-making in cities[25]. Nevertheless, countless other initiatives are also potential candidates. The BBB approach outlines principles rather than specifying particular programs, which should be designed to meet local needs.

It is worth noting that supporting the psychosocial recovery of affected communities has also been identified as essential for BBB[21]. A discussion on psychosocial recovery is, however, outside the scope of this comment.

## BBB through lessons learned from recent events around the world

Estimates[2] suggest that about 500,000 new housing units will be constructed to cater to Turkey's over 1.5 million displaced people. The pathway to meeting these permanent housing needs is not straightforward. Reconstruction following any large disaster is a daunting and massive task, taking not just months or a few years

but decades to reconstruct, as seen in the aftermath of the 2004 Indian Ocean tsunami, 2005 Hurricane Katrina (USA), and the 2010 Haiti earthquake. Technical, environmental, socioeconomic, political, and cultural challenges facing post-disaster recovery efforts may prolong the time for the affected provinces to return to a desired level of normalcy. For example, the 2015 Gorkha, Nepal, earthquake highlighted the negative influence of bureaucratic burdens, egoism, vested interests, and corruption amongst government officials on the post-disaster recovery trajectory[26]. Also, the recovery process in the politically unstable regions of Sri Lanka following the 2004 Indian Ocean tsunami was eight times slower than in areas without conflicts[27]. In many situations, BBB necessitates making trade-offs. For instance, in Nepal, the government provided reconstruction grants after the 2015 Gorkha earthquake, conditional on meeting higher construction standards. Specifically, the government offered hundreds of thousands of homeowners $3000 (£2300) each to rebuild their homes. However, these grants fell short of covering the financial costs of such construction (i.e., they covered only 30-50% of the cost of rebuilding a typical family dwelling), leading to a significant debt burden for many households, further marginalizing the poorest, often in the most precarious housing and financial situations[28,29]. In Haiti, the mandate to build temporary shelters after the 2010 earthquake following BBB principles (e.g., able to withstand Level 3 hurricanes) led to shelters costing significantly more than permanent homes and delayed permanent reconstruction or repair of existing buildings by months/years[30].

We have been investigating in detail the post-disaster recovery and BBB strategies in low- and lower-middle-income countries. We recently concluded a research project to foster a resilient recovery in Central Sulawesi's marginalized communities (Indonesia), with a special focus on school infrastructure. This project included extensive engagement with local and international stakeholders involved in the post-2018 event recovery process[31]. Furthermore, we were part of the UK EEFIT team that visited Palu, Indonesia, in late 2022 to track the recovery process across the affected region[32]. We highlight here some lessons we learned that could be relevant to enhancing BBB in terms of resilience, sustainability, and equity in Turkey, Syria, and other disaster-hit countries.

**Resilience**. TOKİ is a key actor in the post-disaster reconstruction of several thousands of permanent residential buildings made of tunnel-form structural systems across the affected regions in Turkey[2]. In Central Sulawesi, we observed that most initial reconstruction projects in schools, for instance, had better structural integrity and construction quality than projects that started in the later phases of the recovery process. Speaking to stakeholders in Central Sulawesi, we identified that the intensity of quality control exercises faded away as people returned to normalcy. Turkey and Syria should develop legal frameworks and/or policies to ensure seismic code compliance and quality assurance for all new buildings in the affected regions beyond the initial post-disaster reconstruction phase. Moreover, a beneficial strategy for implementing more successful reconstruction plans is to endorse reconstruction programs/activities led by the community, rather than solely focusing on those directed by the owners.

**Sustainability**. According to estimates[33], the 2023 Kahramanmaraş Earthquake Sequence generated 116—210 million tons of debris. In comparison, the 1999 Izmit earthquake generated 13 million tons[34], the 2010 Haiti earthquake generated 19 million tons[35], the 2015 Gorkha earthquake generated four million tons[36], and the 2018 Sulawesi earthquake and tsunami generated about seven million tons[37]. The large debris volume generated from the 2023 Kahramanmaraş Earthquake Sequence can cause

severe economic and environmental issues if not properly managed. Reports[38] have highlighted ongoing poor waste management practices in Turkey—e.g., a lack of waste classification measures for construction and demolition debris. Turkey and Syria have a window of opportunity to recycle the disaster waste to rehabilitate and reconstruct roads and buildings. Apart from meeting local building material demands, this will promote an eco-friendly approach to disaster debris management. Environmental-friendly approaches to disaster waste management have been successful in recent events in other low- and lower-middle-income countries. For example, experience has shown that up to 80-90% of building rubble can be recycled as raw materials for reconstruction[33,39]. Turkey and Syria should ensure that they do not sacrifice effective disaster waste management for the speed of disaster waste removal.

**Social equity**. Engaging all diverse groups (e.g., women, children, youth, persons with disabilities, indigenous peoples, and other marginalized groups) throughout the post-disaster recovery process is also essential. Evidence from the 2018 Palu earthquake shows that a lack of inclusion can impede recovery for different groups, especially socially marginalized groups. For example, there were cases where the government provided permanent housing for groups in an undesired location (i.e., poor proximity to employment location or ancestral land) or the designed units did not offer a desired level of privacy. These housing resettlements were left empty due to unwillingness to relocate. This is not dissimilar to what happened to some Sri Lankan and Samoan communities following the 2004 Indian Ocean tsunami: BBB meant constructing homes far from the coast where communities had their livelihoods. This failed to consider the balance households needed to strike between financial and physical risks. As a result, many of these new homes were abandoned, as people chose to illegally return to the coast despite the risks. Addressing inclusion through appropriate stakeholder engagement with relevant groups from the planning phase can mitigate social issues associated with unsuccessful resettlement programs, which harm livelihood and thus increase vulnerability. Another lesson we learned from the Palu recovery process was the inconsistency in the level of inclusivity in allocating permanent housing units. For example, we observed cases where vulnerable (e.g., aged) people were stuck in temporary shelters for over three years. Turkey and Syria have an opportunity to avoid similar issues by having an inclusive approach to permanent housing allocation.

## Conclusions

Learning from the effects of an earthquake on existing structures, infrastructure, and communities is vital for improving recovery paths. Despite the extensive damage and devastating impact of the 2023 Kahramanmaraş Earthquake Sequence, several inspiring stories emerged. These stories serve as a foundation for a genuine BBB approach, one that is characterized by realistic and achievable goals rather than merely aspirational targets that often remain unfulfilled and divert resources away from essential needs.

BBB is not a one-size-fits-all, and its implementation—often fraught with challenges and complexities—needs to be grounded in the realities of the communities it aims to serve, through a "people-centered" lens. It must consider the voices and needs of those directly affected by the disaster, with residents themselves making the decisions with various levels of ownership and accountability. Every community is unique, with its own challenges, resources, and opportunities. Neglecting the social, cultural, and ethnic aspects of communities during the recovery process can intensify their existing vulnerabilities. Local input, community engagement, and a nuanced understanding of the

social and cultural complexities involved are essential for its successful implementation. The pressure for fast results during recovery can prevent well-intentioned stakeholders (yet with no previous experience in post-disaster environments) from considering community needs.

Furthermore, BBB calls for a holistic and integrated approach, going beyond (only) technical fixes and considering the interconnectedness and interdependencies among different types of risks, such as natural hazards, climate change, conflict, and pandemics. It also calls for a collaborative and coordinated effort among various stakeholders, such as governments, donors, NGOs, the private sector, academia, media, and communities. It requires technical and financial resources, political will, social mobilization, and institutional reform. It is also crucial to consistently monitor recovery efforts, in the short and long term, to ensure adherence to BBB principles and glean insights for enhancing future disaster management strategies.

By embracing these principles, rebuilding what was lost and creating a stronger, more sustainable, and more just future is possible. The path is challenging, but the outcome is a community that has learned from its past, adapted for the present, and prepared for the future.

## Data availability
The data to generate Figs. 1 and 2 are available from the corresponding author, C.G., upon reasonable request.

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

## Acknowledgements
C.G. acknowledges funding from UK Research and Innovation (UKRI) under grant NE/S009000/1, 'Tomorrow's Cities Hub.' E.O. acknowledges funding from UKRI under grant EP/X023710/1, 'MultiVERSE' project, and from the UK Earthquake Engineering Field Investigation Team (EEFIT).

## Author contributions
Both C.G. and E.O. contributed equally to this work. Both authors conceptualized the study and drafted the original manuscript. E.O. collected the data to produce Figs. 1 and 2. Both authors approved the revised version of the manuscript for the final submission.

## Competing interests
The authors declare no competing interests. C.G. is an Editorial Board Member for *Communications Engineering* and was not involved in the editorial review, or the decision to publish, this Article.
