## [Peer Review File · Communications Engineering]

Referee #1 comments

This manuscript covers a very important topic: how do we ensure that societies that experience catastrophe are able to learn from them to ensure more resilient, sustainable and equitable recovery? Great to see this important topic be discussed and highlighted in this way. The paper specifically focuses on the important moment for Turkey and Syria to incorporate 'building back better' principles into their recovery. The paper is structured such that it provides context about the Kahramanmaras earthquake sequence, then covers some of the successful lessons even within those disasters, then describes BBB and then provides specific lessons from the 2018 Palu event (which the authors have studied) which could be relevant to Turkey and Syria. These are important and valuable topics and many important, practical lessons are shared.

While I think the paper is important and valuable for readers, I also provide some feedback around topics that I think could be clarified or deepened, particularly in the discussion of 'building back better'.

Here are some overall comments, followed by some detailed ones:

Overall comments:

The authors could provide more background on "Building Back Better" as a framework for recovery. The narrative would be stronger if the history of BBB in reconstruction was better explained (even briefly). Why was BBB popularised? when? how is it different than what people were doing before?

The section on "an opportunity for BBB" goes over some of the principles of BBB (resilience, sustainability, social equity). It is unclear where these were derived? Are these principles extracted from a UNDRR document?

In addition, it is also unclear how the long list of initiatives that are part of BBB are identified by the authors. For example the section on social equity mentions affordable housing, safe schools, hospitals, job opportunities and microfinances. These are examples of specific programs which could be part of a BBB strategy, but so could innumerable other initiatives. BBB defines principles only, not specific programs, which should be based on local needs. I would suggest restructuring this section to go into more depth around the BBB principles, without defining the programs that make up BBB, or perhaps refer to programs only as possible examples (ideally examples from real post-disaster contexts).

This reviewer believes there is a missed opportunity to provide nuance and highlight the rich literature and even debates around BBB. BBB is not a new concept, and certainly has been central to discussions around reconstruction and recovery ever since the 2004 Indian ocean tsunami. But there are many rich debates on BBB and a lot of nuanced discussion. Here is just a few examples:

- While it is easy to agree with high-level values of building back better, it is much more challenging to agree on how. See paper by Elizabeth Maly entitled 'Rethinking "Build Back Better" in housing reconstruction.'
- BBB often focuses on "what should build back better look like?" but rarely on "who gets to decide what BBB will look like"?

- In the disaster recovery field, there has been push-back against the naivety with which many have promoted “build back better.” Reconstruction following any large disaster is an enormous undertaking even just for meeting basic shelter needs. It takes not months or years but decades to reconstruct (2004 Indian Ocean, Katrina, 2010 Haiti and more just as few examples). Even so, proponents have often claimed that this is also the opportunity to address all kinds of deep-rooted societal ills with little understanding of the complexity involved. Many in the disaster recovery field have therefore argued for reasonable and realistic targets rather than ‘feel-good’ targets which end up being unrealised while also diverting attention and funding away from core needs (See some of work and lectures by Simon Levine, or Elizabeth Maly).
- In many situations, BBB requires compromise. In Nepal, the government provided reconstruction grants conditional on meeting higher construction standards. However these grants were insufficient to meet the financial cost of such construction. As such, the vast majority of household were forced to take out significant loans, and are now burdened with debt. In Haiti, the requirement to build temporary shelters following “BBB” principles meant they were forced to design them to withstand multiple category 3 hurricanes. This led to shelters costing significantly more than permanent homes and delayed permanent reconstruction or repair of existing buildings by months/years.
- Importantly, those defining what BBB should be are rarely those living with the risk. BBB can only be useful if linked to open discussions on ‘acceptable risk,’ involving those who will live with the risk.
- Even while BBB promotes ideas of "social equity," it still primarily focuses on engineering risk reduction. Historically, BBB proponents have failed to recognise the complex risk arbitration that households and communities face. For instance, "building back better" in some Sri-Lankan communities following the 2004 Indian Ocean tsunami meant constructing homes far from the coast where communities had their livelihoods. It failed to recognise the economic risk vs physical risk that households need to balance, and as such it simply led to abandoned homes as people still moved back to the coast.

These are just some of the important discussions around the topic of BBB. I suggest that the authors reference and engage in these discussion.

In the conclusion, the authors miss the opportunity to reconnect the section on “invisible successes amid disaster.” Could be a good place to remind the reader that in fact Turkey has seen examples of many things that worked (e.g. base isolated hospitals), and this can serve as a platform to build from, expand and replicate...

Detailed comments:

Line 38: unclear what “economic incentives” this refers to

Figure 1: circle size depicting displacement seems to only be for those major events in low/middle income countries. Suggest modifying symbols for smaller earthquakes to points so there is no visual misinterpretation of the size of the circles for smaller events.

Section on Invisible Successes in Risk Reduction:

- Suggest referencing Rabonza’s work on the topic, especially as it relates to celebrating success in DRR, and issues of identifying successes amid catastrophe.

Line 93: I think it is meant to say “34 hospital buildings in the affected region” not “34 hospital buildings in the affected building”

Figure 2: some of the data is difficult to see due to the use of solid and overlapping points.

The section “an opportunity for BBB” doesn’t provide a formal definition of BBB, or explain how and when it came to be popular in the disaster recovery and reconstruction field. Providing context would be helpful.

In addition, it is also unclear how the long list of initiatives that are part of BBB are identified by the authors. For example the section on social equity mentions affordable housing, safe schools, hospitals, job opportunities and microfinances. These are examples of specific programs which could be part of a BBB strategy, but so could innumerable other ones. BBB defines principles only, not specific programs, which should be based on local needs.

Several sentences throughout the text mention what Turkey and Syria “must” do or “need to do”. These are very strong words, and are not necessary. Suggest toning down to “should” or even “could,” or “has the opportunity to.”

There are a few switches from general text to a very personal “we”. Suggest adding transitions when this happens to ease the reader into these transitions.

Sentence starting line 233 repeats much of line 227 and using the same words.

The 2023 Kahramanmaraş Earthquake Sequence: A Window of Opportunity for Building a More Resilient, Sustainable, and Equitable Society

Carmine Galasso and Eyitayo A Opabola

We thank the Chief Editor, Rosamund (Ros) Daw, and the anonymous referee for their comments, which helped further improve the quality of the revised manuscript. The comments have been listed below (in *italics*), followed by our responses in purple. Extracts from the manuscript are reported in green, with modifications to the revised manuscript in bold.

Referee comments

This manuscript covers a very important topic: how do we ensure that societies that experience catastrophes are able to learn from them to ensure more resilient, sustainable and equitable recovery? Great to see this important topic be discussed and highlighted in this way. The paper specifically focuses on the important moment for Turkey and Syria to incorporate 'building back better' principles into their recovery. The paper is structured such that it provides context about the Kahramanmaraş earthquake sequence, then covers some of the successful lessons even within those disasters, then describes BBB and then provides specific lessons from the 2018 Palu event (which the authors have studied) which could be relevant to Turkey and Syria. These are important and valuable topics and many important, practical lessons are shared.

While I think the paper is important and valuable for readers, I also provide some feedback around topics that I think could be clarified or deepened, particularly in the discussion of 'building back better'.

We would like to thank the Reviewer for the positive overall assessment of our manuscript. We agree with the comments below on the need to further clarify or deepen some discussion around building back better (BBB). We have attempted to address them, keeping in mind the editorial requirements of the journal, as emphasized by the Chief Editor in their decision letter (*Note that we aim for Commentaries to be concise, so we ask that you keep the manuscript below 3000 words adding only a handful of extra references if necessary.*)

Here are some overall comments, followed by some detailed ones:

Overall comments:

The authors could provide more background on "Building Back Better" as a framework for recovery. The narrative would be stronger if the history of BBB in reconstruction was better explained (even briefly). Why was BBB popularised? when? how is it different than what people were doing before?

Thanks for this comment. The following sentences have been added to Page 4: **"The BBB concept became popular in 2006 during the large-scale reconstruction effort following the Indian Ocean Tsunami disaster in 2004. Ten key principles proposed by former US President Bill Clinton were adopted after that disaster. These principles included a commitment to community-led recovery, promoting fairness and equity, and leaving communities safer by reducing risks and building resilience. Before BBB, post-disaster reconstruction often consisted of simply repairing the physical damage a disaster had induced. However, rebuilding the built environment and infrastructure exactly as they were prior to a disaster often re-created the same vulnerabilities that existed earlier."**

The section on "an opportunity for BBB" goes over some of the principles of BBB (resilience, sustainability, social equity). It is unclear where these were derived? Are these principles extracted from a UNDRR document?

Our starting point was the United Nations Office for Disaster Risk Reduction (UNDRR) definition of BBB, which we now reported in the section "An opportunity to "Build Back Better"" (Page 4). Such a definition mainly highlights the principle of resilience:

"The United Nations Office for Disaster Risk Reduction (UNDRR) defines BBB as "The use of the recovery, rehabilitation and reconstruction phases after a disaster to increase the resilience of nations and communities through integrating disaster risk reduction measures into the restoration of physical infrastructure and societal systems, and into the revitalization of livelihoods, economies and the environment." Based on this definition, the fundamental principle of BBB is resilience."

We have then added our perspectives on the topic, particularly the need to link resilience to sustainability – especially in an era of climate change – and equity. This is now better clarified (Page 4: **“In an era of climate change, BBB should not stop at resilience; it should also embrace sustainability.”**)

We further broadened this section by also discussing the concept of people-centered (housing) recovery, referring to the work of Elizabeth Maly, as suggested by the Reviewer in one of the following comments (Page 5: **“Furthermore, BBB goes beyond just the physical environment; it should also address social equity through a “people-centered” lens”**).

In addition, it is also unclear how the long list of initiatives that are part of BBB are identified by the authors. For example the section on social equity mentions affordable housing, safe schools, hospitals, job opportunities and microfinances. These are examples of specific programs which could be part of a BBB strategy, but so could innumerable other initiatives. BBB defines principles only, not specific programs, which should be based on local needs. I would suggest restructuring this section to go into more depth around the BBB principles, without defining the programs that make up BBB, or perhaps refer to programs only as possible examples (ideally examples from real post-disaster contexts).

We agree with the Reviewer that these are just examples (based on our experience on a number of recent projects in the Global South) of specific programs that could be part of a BBB strategy – but also of more general frameworks for risk-informed urban planning, design, and decision making in cities. The list is obviously not exhaustive.

To reflect the Reviewer’s important comment, we added the following paragraph in our revised manuscript (at the end of the “An opportunity to “Build Back Better” section):

“These are just possible examples (based on our experience on some recent projects in the Global South) of specific programs that could be incorporated into a BBB strategy but also in more general frameworks for risk-informed urban planning, design, and decision making in cities. Nevertheless, countless other initiatives are also potential candidates. The BBB approach outlines principles rather than specifying particular programs, which should be designed to meet local needs.”

This Reviewer believes there is a missed opportunity to provide nuance and highlight the rich literature and even debates around BBB. BBB is not a new concept, and certainly has been central to discussions around reconstruction and recovery ever since the 2004 Indian ocean tsunami. But there are many rich debates on BBB and a lot of nuanced discussion. Here is just a few examples:

- *While it is easy to agree with high-level values of building back better, it is much more challenging to agree on how. See paper by Elizabeth Maly entitled ‘Rethinking “Build Back Better” in housing reconstruction.’*
- *BBB often focuses on “what should build back better look like?” but rarely on “who gets to decide what BBB will look like”?*
- *In the disaster recovery field, there has been push-back against the naivety with which many have promoted “build back better.” Reconstruction following any large disaster is an enormous undertaking even just for meeting basic shelter needs. It takes not months or years but decades to reconstruct (2004 Indian Ocean, Katrina, 2010 Haiti and more just as few examples). Even so, proponents have often claimed that this is also the opportunity to address all kinds of deep-rooted societal ills with little understanding of the complexity involved. Many in the disaster recovery field have therefore argued for reasonable and realistic targets rather than ‘feel-good’ targets which end up being unrealised while also diverting attention and funding away from core needs (See some of work and lectures by Simon Levine, or Elizabeth Maly).*
- *In many situations, BBB requires compromise. In Nepal, the government provided reconstruction grants conditional on meeting higher construction standards. However these grants were insufficient to meet the financial cost of such construction. As such, the vast majority of household were forced to take out significant loans, and are now burdened with debt. In Haiti, the requirement to build temporary shelters following “BBB” principles meant they were forced to design them to withstand multiple category 3 hurricanes. This led to shelters costing significantly more than permanent homes and delayed permanent reconstruction or repair of existing buildings by months/years.*
- *Importantly, those defining what BBB should be are rarely those living with the risk. BBB can only be useful if linked to open discussions on ‘acceptable risk,’ involving those who will live with the risk.*

- Even while BBB promotes ideas of “social equity,” it still primarily focuses on engineering risk reduction. Historically, BBB proponents have failed to recognise the complex risk arbitration that households and communities face. For instance, “building back better” in some Sri-Lankan communities following the 2004 Indian Ocean tsunami meant constructing homes far from the coast where communities had their livelihoods. It failed to recognise the economic risk vs physical risk that households need to balance, and as such it simply led to abandoned homes as people still moved back to the coast.

These are just some of the important discussions around the topic of BBB. I suggest that the authors reference and engage in these discussion.

Thank you for your insightful comments. We appreciate the rich context the Reviewer provided on BBB and their expertise on the topic. We agree that a wealth of literature and nuanced discussion around BBB deserves attention. Unfortunately, as discussed above, we were (and are) constrained by the journal’s editorial requirements in terms of the number of words for our comment piece. We wanted to give space to specific lessons from the 2018 Palu event (which we have studied in detail), which indeed reflect some of the issues raised by the Reviewer and which could be relevant to Turkey and Syria, as discussed in the “BBB through lessons learned from recent events around the world” section.

For instance, we stated:

Page 5 (*An opportunity to “Build Back Better”*): “Furthermore, BBB goes beyond just the physical environment; it should also address social equity **through a “people-centered” lens. In fact, the BBB concept is often too vague to provide clear guidance for housing reconstruction and broader recovery. It is frequently used in a limited sense to denote safer construction or engineering risk reduction without a comprehensive understanding of what constitutes a “better” life for affected individuals, particularly those vulnerable and marginalized.** Disasters often reveal and exacerbate existing inequalities, disproportionately affecting vulnerable populations. It is crucial to ensure that the reconstruction efforts prioritize the needs **of well-informed and empowered residents in decision-making and construction,** promoting inclusivity and reducing disparities”.

Page 5 (*BBB through lessons learned from recent events around the world*): “The pathway to meeting these permanent housing needs is not straightforward. **Reconstruction following any large disaster is a massive undertaking, taking not just months or years but decades to reconstruct, as seen in the aftermath of the 2004 Indian Ocean tsunami, 2005 Hurricane Katrina, and the 2010 Haiti earthquake.** Technical, environmental, socioeconomic, political, and cultural challenges facing post-disaster recovery efforts may prolong the time for the affected provinces to return to a desired level of normalcy”.

Page 6 (*BBB through lessons learned from recent events around the world*): “**In many situations, BBB necessitates making trade-offs. For instance, in Nepal, the government provided reconstruction grants after the 2015 Gorkha earthquake, conditional on meeting higher construction standards. However, these grants fell short of covering the financial costs of such construction, leading to a significant debt burden for many households. In Haiti, the mandate to build temporary shelters following BBB principles led to shelters costing significantly more than permanent homes and delayed permanent reconstruction or repair of existing buildings by months/years**”.

Page 6 (*BBB through lessons learned from recent events around the world*): “For example, there were cases where the government provided permanent housing for groups in an undesired location (i.e., poor proximity to employment location or ancestral land) or the designed units did not offer a desired level of privacy. These housing resettlements were left empty due to unwillingness to relocate. **This is not dissimilar to what happened to some Sri Lankan communities following the 2004 Indian Ocean tsunami: BBB meant constructing homes far from the coast where communities had their livelihoods. This failed to consider the balance households needed to strike between financial and physical risks. As a result, many of these new homes were abandoned, as people chose to return to the coast despite the risks**”.

Page 7 (*Conclusions*): “BBB is not a one-size-fits-all, and its implementation – **often fraught with challenges and complexities – needs to be grounded in the realities of the communities it aims to serve. It must consider the voices and needs of those directly affected by the disaster, with residents themselves making the decisions with various levels of ownership and accountability.** Every community is unique, with its own challenges, resources, and opportunities. Local input, community engagement, and a deep understanding of the social and cultural contexts are essential for successful implementation”.

Note that the sentences in **bold** have been added to the revised manuscript based on the Reviewer's comments. A number of additional references to support these points have been added to the revised manuscript but they are not reported here for brevity.

In the conclusion, the authors miss the opportunity to reconnect the section on “invisible successes amid disaster.” Could be a good place to remind the reader that in fact Turkey has seen examples of many things that worked (e.g. base isolated hospitals), and this can serve as a platform to build from, expand and replicate...

Thanks for this comment. We have now revised the “Conclusions” section to reconnect it to the “Successes in risk reduction measures are often invisible, resulting in a lack of incentives for proactive decision-making.”, as suggested by the Reviewer.

The first paragraph of the “Conclusions” section (Page 7) now reads:

“Learning from the effects of an earthquake on existing structures, infrastructure, and communities is vital for improving recovery paths. Despite the extensive damage and devastating impact of the 2023 Kahramanmaraş Earthquake Sequence, several inspiring stories emerged. These stories serve as a foundation for a genuine BBB approach, one that is characterized by realistic and achievable goals rather than merely aspirational targets that often remain unfulfilled and divert resources away from essential needs.”

Detailed comments:

Line 38: unclear what “economic incentives” this refers to

Thanks for pointing this out. We agree that “economic incentives” was unclear and we replaced it with **“profit motive by multifamily residential building owners or contractors wanting unauthorized extra floors or expanded balconies to maximize profit”** (Page 1).

Figure 1: circle size depicting displacement seems to only be for those major events in low/middle income countries. Suggest modifying symbols for smaller earthquakes to points so there is no visual misinterpretation of the size of the circles for smaller events.

Figure 1 has been modified to reflect the Reviewer's comment. It now appears as:

Figure 1 – Impact of earthquakes on countries in the last two decades. Relationship between earthquake moment magnitude (M_w) and number of fatalities globally in the last two decades. We highlight seven other seismic events in low- and lower-middle-income countries that have experienced earthquakes with $M_w > 6.5$ and a number of fatalities ≥ 4000 with green circles together with the 2023 Turkey-Syria earthquake (red circle). The size of the circles represents the number of displaced people following the eight highlighted events. The grey boxes depict events with a number of fatalities < 4000 or not in the considered country income class. Data were derived from the International Disaster Database (EM-DAT).⁷ The 2018 Central Sulawesi and 2004 Indian Ocean earthquakes triggered tsunamis.

Section on Invisible Successes in Risk Reduction:

- Suggest referencing Rabonza's work on the topic, especially as it relates to celebrating success in DRR, and issues of identifying successes amid catastrophe.

The following references have now been added to the revised manuscript. Thanks for pointing this out.

Rabonza ML, Lin YC, Lallemand D (2022) Learning from success, not catastrophe: Using counterfactual analysis to highlight successful disaster risk reduction interventions. *Frontiers in Earth Science* 10: 847196, <https://doi.org/10.3389/feart.2022.847196>.

Rabonza ML, Lallemand D, Lin YC, Tadepalli S, Wagenaar D, Nguyen M, Choong J, Liu CJN, Sarica GM, Widawati BAM, Balbi M, Khan F, Loos S & Lim TN (2022). Shedding light on avoided disasters: measuring the invisible benefits of disaster risk management using probabilistic counterfactual analysis. *UNDRR Global Assessment Report 2022*, 1-25.

Line 93: I think it is meant to say “34 hospital buildings in the affected region” not “34 hospital buildings in the affected building”

Yes, this has now been fixed. Thanks!

Figure 2: some of the data is difficult to see due to the use of solid and overlapping points.

Figure 2 has now been modified by replacing all filled boxes and circles with empty boxes and circles.

Figure 1 – Impact of seismic isolation systems on post-event functionality of hospitals. Post-event functionality states of 34 hospital buildings in the affected region relative to the observed peak ground accelerations (PGA) derived from the initial USGS shake map. The hospitals were inspected by the United States Earthquake Engineering Research Institute (EERI) team¹⁰ and United Kingdom’s Earthquake Engineering Field Investigation Team (EEFIT)¹⁸ about six weeks after the disaster. There are five base-isolated hospital buildings in the affected region, with another four (not shown in the figure) under construction when the earthquake occurred.

The section “an opportunity for BBB” doesn’t provide a formal definition of BBB, or explain how and when it came to be popular in the disaster recovery and reconstruction field. Providing context would be helpful.

This has now been fixed as discussed above; please see our response to the first two “Overall comments”.

In addition, it is also unclear how the long list of initiatives that are part of BBB are identified by the authors. For example the section on social equity mentions affordable housing, safe schools, hospitals, job opportunities and microfinances. These are examples of specific programs which could be part of a BBB strategy, but so could innumerable other ones. BBB defines principles only, not specific programs, which should be based on local needs.

We agree with the Reviewer that these are just examples (based on our experience on a number of recent projects in the Global South) of specific programs that could be part of a BBB strategy – but also of more general frameworks for risk-informed urban planning, design, and decision making in cities. The list is obviously not exhaustive.

To reflect the Reviewer’s important comment, we added the following paragraph in our revised manuscript (at the end of the “An opportunity to “Build Back Better” section):

“These are just possible examples (based on our experience on some recent projects in the Global South) of specific programs that could be incorporated into a BBB strategy but also in more general

frameworks for risk-informed urban planning, design, and decision making in cities. Nevertheless, countless other initiatives are also potential candidates. The BBB approach outlines principles rather than specifying particular programs, which should be designed to meet local needs.”

Several sentences throughout the text mention what Turkey and Syria “must” do or “need to do”. These are very strong words, and are not necessary. Suggest toning down to “should” or even “could,” or “has the opportunity to.”

Thanks for this comment. This has been fixed in the revised manuscript.

Their are a few switches from general text to a very personal “we”. Suggest adding transitions when this happens to ease the reader into these transitions.

Thanks for this comment. This has been fixed in the revised manuscript.

Sentence starting line 233 repeats much of line 227 and using the same words.

Thanks for this comment. This has been fixed in the revised manuscript and it reads as:

“BBB is not a one-size-fits-all. ~~approach but rather a philosophy that recognises the transformative potential in the aftermath of a disaster.~~ Every community is unique, with its own challenges, resources, and opportunities.”